# Impacts of the Pandemic, Animal Source Food Retailers’ and Consumers’ Knowledge and Attitudes toward COVID-19, and Their Food Safety Practices in Chiang Mai, Thailand

**DOI:** 10.3390/ijerph191610187

**Published:** 2022-08-17

**Authors:** Chalita Jainonthee, Sinh Dang-Xuan, Hung Nguyen-Viet, Fred Unger, Warangkhana Chaisowwong

**Affiliations:** 1Veterinary Public Health and Food Safety Centre for Asia Pacific (VPHCAP), Faculty of Veterinary Medicine, Chiang Mai University, Chiang Mai 50100, Thailand; 2Center of Excellence in Veterinary Public Health, Faculty of Veterinary Medicine, Chiang Mai University, Chiang Mai 50100, Thailand; 3International Livestock Research Institute, Hanoi 100000, Vietnam; 4Department of Veterinary Biosciences and Veterinary Public Health, Faculty of Veterinary Medicine, Chiang Mai University, Chiang Mai 50100, Thailand

**Keywords:** animal source food, consumer, COVID-19, food, KAP, meat, retailer, Thailand

## Abstract

The COVID-19 pandemic affected the food supply chain, retailers, and consumers owing to infection awareness. This study evaluated the impacts COVID-19 on ASF retailers’ businesses and consumers’ livelihoods, as well as their knowledge toward the disease, attitudes, and food safety practices to prevent infections. The study includes a cross-sectional component that was conducted in urban/peri-urban (U/PU) and rural areas in Chiang Mai province. In another part of the study, a structured questionnaire was developed for animal source food (ASF) retailers and consumers, with three primary parts for data analysis: general information, COVID-19 impacts, and knowledge, attitudes, and practices (KAP) assessment. Data corresponding to three periods of interest (before the COVID-19 outbreak, during partial lockdown, and present) were gathered and analyzed. In this study, 155 retailers and 150 consumers participated, of which the majority of the respondents were female (70.3% and 82.7%, respectively) with average ages of 47.4 and 44.9 years, respectively. The most noticeable effect of COVID-19 was a decline in income for retailers and consumers. The KAP scores of consumers in both areas were not significantly different, whereas the retailer attitudes toward COVID-19 prevention and food safety practices scored more highly in rural areas than in U/PU. During the partial lockdown, food safety practices significantly improved relative to the time preceding the outbreak, and these practices have remained constant to the present day. The results revealed that gender, age group, business type, and type of ASF retailers were associated with the KAP of the retailers, whereas gender, age group, education, number of family members, and occupation were associated with the KAP of the consumers. Our findings provide in-depth information about the effects of COVID-19 on ASF retailers and consumers, as well as their KAP regarding the outbreak and food safety, which may serve as support in developing policies for improved health and food safety.

## 1. Introduction

The COVID-19 pandemic is continuing to have considerable impacts on public health, way of life, and economics globally. Knowing that coronaviruses originate from animals, we hypothesized that SARS-CoV-2 would be human-transmissible [1]. The outbreak demonstrates that the animal–human interface has the potential to serve as the primary source of emerging zoonotic diseases [2]. Bats have been identified as the natural host of these viruses, and infections in human were initiated from the virus crossing species barriers via potential intermediate hosts [3,4,5,6]. As for the virus being novel, little was known about it in the first phase of the outbreak. There were reports at the time of an outbreak that was linked to a Chinese wholesale food market. A human case was reported to have tested positive for COVID-19 without evidence of wildlife trading on the market, and the virus was detected on salmon-importing chopping boards [7,8]. During COVID-19, food safety was a major topic of discussion [9]. Despite the lack of scientific proof implicating salmon as the source of the sickness, people feared that salmon and other commonly consumed meats, known as animal source food (ASF), could also be potential sources of infection.

After the spread of infections globally as a pandemic, COVID-19 national containment measures that were implemented by health agencies and governments involved lockdown of communes, cities, and even provinces. Aligned measures of movement restrictions, included closing of public venues such as markets and schools, isolating infected people or quarantining suspected cases, and social distancing were implemented to limit the number of new infections [10,11,12]. Additionally, an important aspect for governments and authorities is to ensure adequate and safe food sources, especially in infected areas (lanes, apartments, communes, and cities), or when heavy lockdown and restrictions are applied [13,14]. Therefore, food production and business activities, as well as food safety practices and consumption, especially relating to ASF, e.g., pork, chicken, egg, and milk, will likely be affected. The lockdown restrictions have impacted meat price and supply due to lower production and limited access for consumers [15]. In addition to demand-side shocks from measurement implications, supply-side disruptions have been addressed. Lockdown measures disrupted transportation networks, while infections of workers in meat processing plants caused problems in the food supply chain as a result of labor shortages [16,17]. Consequently, consumers are concerned about the possibility of cross-contamination of food by a virus due to the infection of food business operators. These impacts have not only caused undesirable effects in terms of economics but also changed the food safety practice of retailers due to consumer fears of unsafe ASF. 

Thailand is renowned for its ability to provide ASF to feed local communities, primarily pork, chicken, fish and seafood, and eggs, and is one of the world’s leading exporters of chicken meat. According to the COVID-19 risk assessment, private sector businesses and government offices were categorized into four groups based on the number of individuals that were present, type of interaction, and duration of contact [18]. Fresh markets were categorized as the second-highest level of risk or intermediate-risk businesses, after crowded places namely exhibition centers and theaters, that required stringent physical distance regulations. To the best of our knowledge, knowledge and attitudes regarding COVID-19 have been evaluated in a variety of population groups and nations around the world, including health professionals, students, travelers, and the general public [19,20,21,22,23,24,25,26,27]. However, little is known about the knowledge and attitudes of ASF retailers regarding the COVID-19 pandemic and its impact on their food safety practices. Even though consumers’ food selection, cooking, and eating habits that were affected by the pandemic were evaluated [28,29,30], it is obvious that there is a gap of ASF-related food safety practices, which is the subject of the current study.

The aims of the present study were to investigate the impacts of the COVID-19 pandemic and to evaluate the disease knowledge, attitudes, and practices (KAP) in food safety of Thai ASF retailers and consumers before, during, and after the initial phase of the outbreak. We selected Chiang Mai province as our study area in order to determine whether there is a difference in the KAP between the urban/peri-urban (U/PU) and rural populations among retailers and consumers. Regarding the impacts of COVID-19, ASF retailers’ businesses and consumers’ livelihood were assessed. Due to the emergence of human infectious diseases, research findings may serve as proof of infection awareness and improvements in food safety practices for maintaining food quality and reducing pathogen cross-contamination. These are also beneficial to policy-makers and other stakeholders in order to establish effective solutions and risk communication with ASF retailers and consumers.

## 2. Materials and Methods

### 2.1. Study Area

This study was conducted in Chiang Mai province, Thailand. Chiang Mai is geographically located in the northern region of Thailand and is the province with the highest population within this region. Chiang Mai is also a business center of the region where animal source food (ASF) is mainly distributed to other provinces in the region. Thus, it was selected as the location to study the impacts of the COVID-19 pandemic on ASF retailers and consumers. The survey was mainly conducted in two geographic areas, defined according to residence density as representing either U/PU or rural context, in order to determine whether COVID-19 impacts and KAP of respondents vary between geographic areas. The area selection of U/PU was based on the definition where administration authority belongs to municipality. The municipal areas that were selected for this study have more than 10,000 residents, whereas rural areas are defined by the areas that are managed by a local administrative organization and mostly have a lower number of residents than U/PU areas. 

In the U/PU areas, the selected subdistricts include Chang Khlan, Chang Phueak, Hai Ya, Suthep, Mae Hia, Pa Daet, Pa Tan, San Phisuea, Si Phum, Nong Phueng, Tha Wang Tan, San Kamphaeng, Ton Pao, Nong Chom, Nong Han, San Na Meng, San Sai Noi, and San Pu Loei. In the rural areas, the selected subdistricts include Hang Dong, Nam Phrae, Ban Klang, Thung Satok, Song Khwae, Samoeng Tai, Ban Kad, Don Pao, Ban Luang, Chai Sathan, Yang Noeng, Huai Sai, San Klang, Mae Pu Kha, San Phranet, San Sai Luang, Choeng Doi, Mae Khue, Mae Pang, Mae Waen, Wiang, Ban Sa Ha Khon, Khi Lek, Mae Sa, Rim Nuea, Rim Tai, Ban Chang, Inthakhin, Mae Ho Phra, San Maha Phon, and Chiang Dao (Figure 1). The selection of subdistricts for this study was based on convenience sampling. The target study populations were ASF retailers and household consumers in both areas. Interviews took place in a fresh market within the selected area.

### 2.2. Study Design and Sample Size

A cross-sectional design was applied to collect data from participants using structured questionnaires to assess the impacts of the COVID-19 pandemic on ASF retailers and consumers. Separate questionnaires were designed for ASF retailers and consumers. The questionnaire was divided into three main parts: (1) general information of participants, (2) impacts of COVID-19, and (3) KAP assessment. Interviews to determine the impacts of COVID-19 and practices toward food safety were structured and divided according to three periods of time: before the COVID-19 outbreak (before February 2020), during partial lockdown (several weeks in April 2020), and present (from May 2020 until the time of interview).

The study was designed to include at least a total of 300 respondents consisting of 150 retailers (75 retailers each context) and 150 consumers (75 consumers each context). The sample size calculation for each value chain actor (retailer and consumer) was based on comparison of the two proportions. At least one key question with an expected difference between U/PU and rural context was used for sample size calculation. For the consumer, the key question was “frequency of washing and disinfection hands and kitchen utensils practices” in U/PU areas (assuming of *p* = 80%) compared with rural areas (assuming *p* = 55%). For the retailer group, the key question was the “effect of COVID-19 to their ASF selling” for U/PU and rural retailers an expected reduction of 45% and 20%, respectively. Using intra-cluster correlation in a sampling area of 0.1, the average cluster size of 5, the required sample size for consumers and retailers in each area was 75 participants. The number of retailers based on type of ASF was adjustable depending on the retailer respondents participating in the study. Additional interviews with retailers and consumers were also conducted to compensate for incomplete questionnaires. Only complete questionnaires were included in the statistical analysis. In the present field survey, a total of 305 respondents participated in the questionnaire interviews, of which 155 participants were ASF retailers and 150 participants were consumers. Questionnaire interviews were conducted from December 2020 to March 2021. The scope of the study focuses on the impacts and changes in the practices of retailers and consumers due to the COVID-19 pandemic. The questionnaire survey scope covers the impacts of the outbreak from the first wave of COVID-19 in Thailand, which began in early March 2020, and the partial lockdown was implemented in April 2020 [23].

### 2.3. Selection Criterion of Participants

ASF retailers are defined as persons who sell their meat products in fresh markets in the selected areas. The selection of ASF retailers was mainly based on their agreeing to participate in the study. The selection of ASF retailers would cover all kinds of retailers (pork, poultry, beef, and fish/seafood) if applicable. Consumers are defined as the person with the main responsibility for purchasing and preparing ASF in the household. Additionally, a consumer must be over 18 years old and their house is within a 5 km radius from the market. 

### 2.4. Ethics Statement

Ethical approval for the study was granted through the International Livestock Research Institute (No. ILRI-IREC2020-35) and the Office of Research Ethics, Research Institute for Health Sciences, Chiang Mai University (No. 24/63). Prospective participants in the study were briefed about how their responses would be used and informed that they could withdrawal from the study at any time. Those who chose to proceed with the study were asked to review and sign a consent form.

### 2.5. Data Collection

The selection of ASF retailers and consumers was based on their agreeing to participate in the study. The staff read the study objectives to the participant and asked them to sign their consent before interviewing. The participant responses were gathered using structured questionnaires. The research team developed questionnaires in English, which were then translated into Thai for the collection of response information. Questionnaires were revised for validation under the supervision of senior researchers. The retailer questionnaire consisted of four sections (Appendix A): participant information (section A, 7 items); impact of COVID-19 on ASF business and income (section B, 10 items); KAP on COVID-19 and prevention measures (section C, 4 items); and food safety practices and behaviors (section D, 2 items). In an evaluation of retailers’ knowledge of COVID-19, responses to questions C1 through C3 were scored, while questions C4 and D2 assessed retailers’ attitudes. Additionally, the response to question D1 concerning food safety practices was assessed. There were 5 sections in the consumer questionnaire (Appendix A): participant information (section A, 8 items); impact of COVID-19 to ASF purchasing and selection (section B, 5 items); impact of COVID-19 to consumer’s job and livelihood (section C, 5 items); KAP on COVID-19 and prevention measures (section D, 4 items); and food safety practices and behaviors (section E, 2 items). In an evaluation of consumers’ knowledge of COVID-19, responses to questions D1 through D3 were scored, while questions D4 and E2 assessed the consumers’ attitudes. Additionally, the response to question E1 concerning food safety practices was assessed. 

Interviewing of the participants took place in the markets where ASF retailers were selling their meat products and consumers were purchasing these meat products. Interviews were conducted in Thai and lasted approximately 30 min for each individual. The data were recorded in the questionnaires and then translated back into English, and the information was entered into an online database. 

### 2.6. Data Management and Analysis

All the responses were exported to Microsoft Excel spreadsheets for cleaning, processing, and further analysis. Descriptive statistical analysis of the general information, impacts of COVID-19, and KAP of the retailers and consumers was conducted. Each validated question in the knowledge assessment section was independently analyzed with answers assigned a score of either 1 (correct) or 0 (incorrect). Additionally, for knowledge assessment according to COVID-19 prevention and control, the response of each question was given a scale rating based on expert opinion. To analyze the attitude and practice of each individual, the response for each question was scored according to a point scale, where higher scores indicate a more positive attitude or practice. To analyze how individual participants performed in each of the knowledge, attitude, and practice categories overall, the sum of each participant’s answers for that section was calculated. The mean scores of the groups of retailers or consumers were then calculated. The respondents with knowledge, attitude, and practice scores of equal or greater than the mean scores were considered to have good knowledge, attitudes, and practices, while those who had scores below the mean were categorized as having poor knowledge, attitudes, and practices. To assess the KAP scores between two areas, an independent *t*-test was separately performed for knowledge toward COVID-19, attitudes toward COVID-19 prevention, attitudes toward food safety practice, and practices toward food safety of the retailers and consumers. Additionally, a paired *t*-test was performed to analyze the differences in the practices of the retailers and consumers in separate areas and for comparing between the periods before the COVID-19 outbreak, during partial lockdown, and present. Univariable analysis (chi-square or Fisher’s exact test, where appropriate) was used to determine the factors that were associated with the observed categories. Factors with *p* ≤ 0.20 were included in the multivariable analysis using multivariable logistic regression to determine the association between potential factors and the level of knowledge, attitudes, and practices. Data analyses were performed using IBM SPSS Statistics version 26 (Armonk, New York, NY, USA). *p* < 0.05 was considered to indicate statistically significant differences.

## 3. Results

### 3.1. Participant Demographics

A total of 305 respondents participated in the structured interviews, of which 155 participants were ASF retailers and 150 participants were consumers. Most the interviewed ASF retailers and consumers were females (70.3% and 82.7%, respectively). The data were collected from 72 and 83 ASF retailers from U/PU and rural areas, respectively. The average ages of ASF retailers in U/PU and rural areas were 46.8 and 48 years old, respectively. ASF retailers in both areas have mainly achieved a primary level of education, with 40.3% in U/PU and 41% in rural areas, respectively. The business type of participants in both areas were mostly retail (73.6% in U/PU and 85.5% in rural areas, respectively). Fish/seafood was the most popular type of meat that was regularly sold by retailers in U/PU areas (43%), while pork was the main type of meat that was mostly sold by retailers in rural areas (43.4%). Beef was the least popular meat that was sold in both types of areas according to the number of the retailers that participated in this study.

For consumers, data were collected from 75 participants in each area. Most of the participants were female (85.3% and 80% in U/PU and rural areas, respectively). The average age of consumers in U/PU areas was 44.9 years and 50.9 years in rural areas. Most U/PU consumers have achieved college or higher education (38.7%), while most rural participants have achieved primary education (48%). Most of the consumers were running their own business as their main occupation (89.3% and 78.7% in U/PU and rural areas, respectively). In U/PU areas, most participants had a household monthly income between USD 200 and 400 (28%) while rural household monthly incomes were between either USD 200 and 400 or USD 400 and 600 (24% for each). The number of family members of the consumers ranged between 3 and 5 persons (50.7% and 57.3% in U/PU and rural areas, respectively). The demographic characteristics of ASF retailers and consumers are shown in Table 1.

### 3.2. Impacts of COVID-19 to Retailers

#### 3.2.1. Average Amount of ASF Sold at Different Periods

The daily beef sales volumes of the two areas were significantly different. In rural areas, the amount of beef that was sold was high, at 112 kg/day, while in U/PU areas there was an average of 36.3 kg/day, which corresponds to the meat type that had the least demand. In U/PU areas, up to 40% of poultry and fish/seafood were sold less frequently during partial lockdown, while the amount of pork sold decreased 34%. After lockdown, there was a slight increase in the amount of ASF that was sold but not to the levels that were seen prior to the COVID-19 outbreak (Appendix A). In rural areas, the amounts of poultry and beef that were sold during partial lockdown had largely decreased, by up to 60% compared with the time before the outbreak. During lockdown, the amounts of pork and fish/seafood decreased 20% and 33.5%, respectively. Differently, in rural areas, supplies of all types of ASF have been continually decreasing from the start of partial lockdown until present (Appendix A).

#### 3.2.2. ASF Suppliers

The main suppliers of ASF in U/PU and rural areas were wholesalers/middlemen. Most retailers usually purchased ASF from the suppliers to sell on a daily basis. ASF products were less supplied by farmers, slaughterhouses, self-sufficient, and integrated companies (Appendix A). Over 80% of retailers in both areas did not face difficulty of ASF seeking from their suppliers during partial lockdown and at present (87.5% during partial lockdown and 91.7% at present in U/PU areas, 86.7% and 91.6%, respectively, in rural areas) compared with the time before the outbreak (Appendix A).

#### 3.2.3. Average Numbers of ASF Shops in the Same Market/Area

In U/PU areas, fish/seafood shops were the most popular type of ASF, with an average number of 11 shops in the same areas, followed by poultry (eight shops), pork (four shops at the time before the COVID-19 outbreak and three after the outbreak), and beef (two shops). In rural areas, there were the most pork shops (five shops) in the same market or area compared to poultry (two shops), beef (one shop), and fish/seafood shops (two shops). AS an overview, the number of ASF shops were not remarkably changed after the COVID-19 outbreak in both of the study areas (Appendix A).

#### 3.2.4. Frequency of Selling and ASF Distributions

Markets remained the most popular selling location for retailers in both areas prior to the COVID-19 outbreak, during the partial lockdown and at the time of interviewing. ASF was sold daily by up to 96% of retailers in U/PU and 81% in rural areas. The COVID-19 pandemic had little effect on the selling frequency. A smaller percentage of retailers also offered their ASF products in other locations, such as flea markets and their own stores outside of markets, online, or through distribution to other locations, such as restaurants (Appendix A).

#### 3.2.5. Labor Resources and Time Spent on Selling ASF 

There were the same results for labor resources and time spent on selling ASF in both U/PU and rural areas. The retailers usually sold ASF by themselves or had one assistant to help with selling. As an overview, it took about 9–10 h of selling per day (full-time job). The time that was spent on selling was not affected by the outbreak of COVID-19 (Appendix A).

#### 3.2.6. Impacts of COVID-19 to Retailer’s Income

A total of 76% of retailers in U/PU areas had reduced income during partial lockdown (52% decrease in income compared with before the outbreak), while 5.3% had increased income (40% increase in income compared with before the outbreak). They continued to have outbreak-related effects following lockdown although to a lesser extent than during the partial lockdown period. The reasons for decreased income among those who were affected included buyers purchasing less than they did previously (42.6%), buyers were less likely to go to the market (38.6%), markets were closed due to the outbreak (8.9%), retailers could not find ASF products to sell (3%), and other (6.9%), such as buyers feeling uneasy about visiting the market (fearful of infection) and buyers’ income had decreased. The increased revenue of retailers was attributed to an increase in number of customers (42.9% of retailers) and government-supported expenditure (28.6%), in addition to customers purchasing more ASF in order to store their food, corresponding to an increase in selling hours per day (14.3% each) (Figure 2A).

In rural areas, 65.1% of retailers had dropped their income during partial lockdown (50.8% of income was decreased from before the COVID-19 outbreak), while 3.6% had raised their income (50% of income was increased). At the time of interviewing, they were still feeling the effects of the pandemic, albeit to a lesser extent than they were during the partial lockdown. The reasons for decreased income among those retailers that were affected by the COVID-19 pandemic included buyers were less likely to go to the market (45.9%), buyers purchasing less than they did previously (37.6%), markets were closed due to the outbreak (8.2%), retailers could not find ASF products to sell (2.4%), and other (5.9%), such as buyers seeking alternative cheaper food types and buyers’ income having decreased. The increased revenue of some retailers was due to customers purchasing more ASF in order to store the food (57.1%), increase in the number of customers (28.6%), and customers had purchased more of a certain type of ASF because it was cheaper than another (14.3%) (Figure 2B).

### 3.3. Impacts of COVID-19 on Consumers

#### 3.3.1. Impacts of COVID-19 on ASF Purchasing and Selection

In U/PU areas, consumers were mostly liked to buy ASF one to three times a week. Some consumers purchased ASF on a daily basis. The same trend of ASF purchase was observed in rural areas. In both areas, the purchasing frequency was not statistically significantly different during partial lockdown and at present compared with the time before the COVID-19 outbreak (Appendix A).

#### 3.3.2. Preference Type of Retails and Purchasing Frequency for Household Consumption

In both U/PU and rural areas, traditional markets were popular sources for purchasing ASF. In rural areas, the majority of consumers purchased ASF from traditional or fresh markets at least four times per week, but in U/PU areas, the majority of consumers preferred to purchase ASF on a daily basis, and others had purchased ASF two to three times per week. Additional retail options, such as flea markets, supermarkets, convenience stores, street vendors, and online delivery, were available to some consumers in both areas (Appendix A).

#### 3.3.3. Average Amount of ASF Purchasing per Shopping Trip

In U/PU areas, the number of purchases has not changed significantly as a result of the COVID-19 outbreak. Consumers continued to purchase each variety of ASF at the same amount as they did prior to the outbreak. Fish/seafood was the preferred type of ASF that was consumed in U/PU areas based on purchased amounts of 3.2 and 2.8 kg prior to the outbreak and during partial lockdown, respectively. In rural areas, pork was the main type of ASF that was purchased by consumers. However, the amount of pork that was purchased per shopping trip was reduced by a remarkable 37% since the COVID-19 outbreak (from 2.7 kg prior to the COVID-19 outbreak compared with 1.7 kg during partial lockdown), but the demand for other types of ASF remained constant (Appendix A). Additionally, during partial lockdown, eggs were also a preferred type of ASF, with 34 and 24 eggs purchased per shopping trip in U/PU and rural areas, respectively. However, wildlife was not a preferred type of ASF for consumers in both areas, with the lowest amount of consumption effectively zero in all three periods (97% and 100% of consumers in U/PU and rural areas, respectively).

#### 3.3.4. Difficulties in Purchasing ASF at Different Periods

There was no change in the desire for and purchase of ASF in either U/PU or rural areas both during partial lockdown and at present compared with prior to the COVID-19 outbreak. However, most consumers lacked experience in purchasing beef and wildlife meat, with the majority of responses indicating that they did not know about difficulties in purchasing those types of meat (Appendix A).

#### 3.3.5. Impacts of COVID-19 to Consumer’s Incomes and Livelihood

More than 90% of the consumers in both areas were affected by the COVID-19 outbreak (91% and 93% in U/PU and rural areas, respectively). The consumers’ incomes were halved (43.5% of affected consumers) or reduced by 10–40% (31.9% of affected consumers) in U/PU areas. Additionally, COVID-19 reduced consumers’ income by almost half (60% of affected consumers) and by 60–90% (28.6% of affected consumers) in rural areas. The primary cause of consumers’ income loss in both areas was a reduction in the number of clients in their businesses/jobs (52.9% and 70% of the affected consumers in U/PU and rural areas, respectively) followed by the enforced lockdown by the government (32.4% and 18.6% of the affected consumers in U/PU and rural areas, respectively). Only one consumer (1.3%) in U/PU areas had benefitted from the outbreak due to the increased number of customers staying at home (Figure 3). 

The majority of consumers in both areas perceived more negative changes in their livelihoods than they did prior to the COVID-19 outbreak. During partial lockdown and at present, consumers in U/PU areas faced difficulties that were related to not having enough food to eat (30.7% and 25.3% increase, respectively) and being unable to eat healthy and nutritious food (20% and 14.7% increase, respectively) due to a lack of money or other resources. Furthermore, other impacts on livelihood included consumers eating only a few kinds of foods or were hungry but did not eat because there was not enough money or other resources for food. In rural areas, consumers felt worried about not having enough food to eat because of a lack of money or other resources (24% and 21.3% increased, respectively) compared with before the outbreak. Other impacts on livelihood included consumers being unable to eat healthy and nutritious food or eating only a few kinds of foods because of lack of money or other resources that were needed to obtain food (Appendix A).

### 3.4. KAP of Retailers toward COVID-19 and Food Safety

#### 3.4.1. Knowledge of ASF Retailers toward COVID-19

Most of the retailers in both areas were aware that the COVID-19 causative disease agent was viral infection (68.1% in U/PU and 62.7% in rural areas), while about the one-third of the retailers had heard about COVID-19 before but did not know about the causative agent. Regarding their knowledge, the retailers agreed that the main ways of COVID-19 transmission were from a person who is sneezing and coughing (94.4% and 97.6% in U/PU and rural areas, respectively) and from touching surfaces or objects that were covered with respiratory droplets (86.1% and 95.2% in U/PU and rural areas, respectively). Airborne transmission was raised as a secondary means of COVID-19 transmission (each of 63.9% in both areas). Furthermore, most did not agree that COVID-19 was transmitted by eating livestock (80.6% and 81.9% in U/PU and rural areas, respectively), through vectors (65.3% and 69.9% in U/PU and rural areas, respectively), and through direct contact with animals (43.1% and 53% in U/PU and rural areas, respectively) (Appendix A).

Almost 100% of the retailers in both areas agreed that the most important means to stop disease spreading were frequently washing hands, covering mouth and nose while sneezing and coughing, avoiding contact with people when having any symptoms, wearing a mask and practicing social distancing when going out of the house, and keeping people in quarantine after they returned from risky areas or if they had been in contact with an infected person. In addition, only staying at home (75% and 80.7% in U/PU and rural areas, respectively) and avoiding meeting with strangers (76.4% and 77.1% in U/PU and rural areas, respectively) were raised as secondary important measures. Regarding their knowledge, avoiding eating any ASF and raw products, such as vegetables, were not important for disease prevention. However, about half thought that wildlife could be the source of infection, while the rest thought that it was not important to avoid eating wildlife in order to prevent disease spread (Appendix A).

#### 3.4.2. Attitudes of Retailers toward COVID-19 Prevention and Food Safety Practices

Regarding the attitudes of retailers toward COVID-19 prevention, frequently washing hands, covering mouth and nose while sneezing and coughing, avoiding contact with people when having any symptoms, wearing a mask when going out of the house, and practicing social distancing when going out of the house were raised as the most important preventive measures in both areas (almost 100% of each item). Avoiding eating wildlife was a secondary practice for the participants to stop the disease by themselves (68.1% and 78.3% in U/PU and rural areas, respectively). The retailers perceived that avoiding eating any ASF was not an effective way to stop COVID-19 infection (63.9% and 74.7% in U/PU and rural areas, respectively). Additionally, in rural areas, their attitudes on the measures of only staying at home and avoiding meeting with strangers were considered non-applicable practices. However, in U/PU areas, the retailers did agree with those measures (34.7% and 36.1%, respectively), while approximately the same proportion of them did not agree that those measures were applicable (Appendix A).

Regarding food safety perception and behavior, most retailers agreed that washing/cleaning and disinfecting shop equipment and facilities, washing and disinfecting hands during selling ASF, and wearing a mask were necessary to reduce health risks and ensure that food is safe for consumption (above 80% of each item in both areas). In rural areas, the retailers realized that food inspection by authorities and seller health checks were also necessary for food safety (89.2% and 80.7%, respectively) compared with those in U/PU areas, who were less concerned (70.8% and 68.1%, respectively). The retailers disagreed that consumers were concerned about their health and not happy when they were wearing mask and gloves (38.9% in U/PU and 62.7% in rural areas) (Appendix A).

#### 3.4.3. Food Safety Practices of Retailers

In U/PU areas, washing/cleaning the ASF shop, keeping fresh ASF in a cooling facility or on ice, eating/drinking at the shop, and washing hands were practices that were carried out frequently by retailers before the COVID-19 outbreak, whereas wearing a mask and disinfecting hands were less frequent practices from before. Following the introduction of COVID-19 into the areas, disinfection of hands and wearing a mask were two habits that significantly changed. Furthermore, since the partial lockdown until present, retailers have increased their frequency of washing/cleaning and disinfecting their stalls or shops, washing hands, wearing gloves, having ASF quality checks by authorities, and keeping fresh ASF in a cooling facility or on ice. However, they retained an adverse practice of eating/drinking at the shop (Appendix A).

In rural areas, keeping fresh ASF in a cooling facility or on ice, washing/cleaning stalls or shops, and washing hands were frequent practices of retailers before the COVID-19 outbreak. Most retailers had never practiced ASF quality checking, disinfecting of hands, and wearing gloves and a mask before the outbreak. However, keeping fresh ASF in a cooling facility and eating/drinking at the shop have been a consistent practice up to the present. Additionally, after the outbreak, retailers changed their practices through the increased frequency of having ASF quality checks by authorities, washing and disinfecting of hands, and washing/cleaning and disinfecting of stalls or shops. However, only some changed their practices by wearing gloves and a mask when selling ASF (10.8% increased) (Appendix A).

#### 3.4.4. Factors Related to KAP on COVID-19 and Food Safety of Retailers

Independent *t*-testing showed that there were no statistically significant differences in the retailers’ knowledge toward COVID-19 transmission and prevention measures and food safety practices between the two areas. Regarding retailers’ attitudes, the rural respondents showed significantly higher scores for attitudes toward COVID-19 prevention and attitudes toward food safety practices than the U/PU group (Appendix A). In comparison to the practices of retailers prior to the COVID-19 outbreak, the practice scores during partial lockdown and at present were observed to be significantly higher in both areas. However, practices of the retailers in both areas during partial lockdown compared to at present were not statistically significantly different (Appendix A).

The results from univariable analysis of knowledge indicated that 75% of the retailers had knowledge about COVID-19 transmission and prevention (Table 2). The univariable test of gender and knowledge of COVID-19 showed that the odds ratio (OR) between male and female retailers was 2.69, indicating female retailers were more likely to have better knowledge of COVID-19 than males (95%CI = 1.26, 5.76). Overall, 67% of the retailers had favorable attitudes toward COVID-19 prevention, while 59% of them had positive attitudes toward food safety practices. Regarding the attitudes toward food safety practices, retailers who had only a retail business type were found to have more positive attitudes than those who operated both wholesale and retail businesses (OR = 0.23, 95%CI = 0.10, 0.54).

Considering the time during partial lockdown, above half (55%) of the retailers performed good practices toward food safety. Retailers in the age group of 30–39 were more likely to have the best practices toward food safety among other groups (OR = 3.75, 95%CI = 1.11, 12.62), and the age group 20–29 was the worst performing group. Regarding the type of ASF, fish/seafood retailers were less likely to perform good practices toward food safety than other ASF retailers (OR = 0.31, 95%CI = 0.14, 0.68).

The results from multivariable analysis illustrate that gender was a factor that was related to knowledge toward COVID-19 of the retailers (Appendix A). There was no factor that was significantly related to attitudes toward COVID-19 prevention. Business type was the only factor that had *p* ≤ 0.20 and was included in multivariable analysis, with the same results, as shown in Table 2. Furthermore, age group and type of ASF were factors that were significantly related to practices toward food safety of the retailers.

### 3.5. KAP of Consumers toward COVID-19 and Food Safety

#### 3.5.1. Knowledge of Consumers toward COVID-19

Above 70% of the consumers (78.7% in U/PU and 70.7% in rural areas) have knowledge about the COVID-19 causative agent. Most consumers in both areas had a similar tendency of answers regarding the transmission of COVID-19. They had knowledge about the route of transmission of the viruses being mainly through air, from a person who is sneezing and coughing, and from touching surfaces or objects that were covered with respiratory droplets of infected persons. Regarding their knowledge, the viruses are less likely to be transmitted through vectors and from eating livestock products. However, the majority of consumers reflected that COVID-19 infections were not from eating wildlife products (40% in U/PU and 44% in rural areas), while almost half of them thought that infections were more likely or possibly occurred from eating those products (total of 49.3% and 42.6% in U/PU and rural areas, respectively). Almost half of the consumers agreed that it is not possible for COVID-19 viruses to be transferred from contact with healthy people (48% in U/PU and 44% in rural areas), while some (34.7% and 33.3% of consumers in U/PU and rural areas, respectively) thought that it is possible for healthy individuals to pass the viruses to others (Appendix A).

From interviewing, nearly 100% of the consumers in U/PU and rural areas have given answers regarding significant responses for the prevention and control of COVID-19. Frequently washing hands, covering mouth and nose when sneezing and coughing, avoiding contact with people when having any symptoms, wearing a mask and practicing social distancing when going out of the house, and keeping people in quarantine after they have returned from risky areas or been in contact with an infected person were raised as the most important means to prevent the spread of disease. Additionally, only staying at home was also raised as an important measure to avoid spreading of the disease in rural areas (92%) and to avoid meeting with strangers was raised as a secondary preventive measure (74.7%), while in U/PU, only staying at home and avoiding meeting with strangers were raised as secondary preventive measures of COVID-19 (77.3% and 76%, respectively). However, most consumers in rural areas thought that avoiding eating products, such as wildlife, raw products (vegetable), or any ASF, were not important in disease control (Appendix A).

#### 3.5.2. Attitudes of Consumers toward COVID-19 Prevention and Food Safety Practices

In the opinion of consumers, preventive measures, including frequently washing hands, covering mouth and nose while sneezing and coughing, avoiding contact with people when having any symptoms, wearing a mask, and practicing social distancing when going out of the house were the most effective measures for COVID-19 prevention (100% of all items in U/PU areas and nearly 100% of all items in rural areas). Avoiding eating wildlife was a secondary preventive means that the consumers in both areas thought they could do to stop the disease spreading (70.7% and 76% in U/PU and rural areas, respectively). However, only staying at home, avoiding meeting with strangers, and avoiding eating any ASF and raw products were regarded as less effective in controlling COVID-19 spread (Appendix A).

Regarding their attitudes toward food safety practices, the majority of consumers in both areas strongly agreed that looking for a shop with good practices to buy ASF, expecting sellers to use masks and gloves when selling ASF, and frequently washing or cleaning kitchen equipment helps to reduce health risks, while frequently washing and disinfecting hands, food quality and safety inspections by authorities, and avoiding raw or undercooked ASF are all necessary. However, as a result of their positive attitudes, they did not believe that shop owners wearing masks and gloves while selling were concealing health problems (Appendix A).

#### 3.5.3. Food Safety Practices of Consumers

In U/PU areas, the greatest change in consumer behavior after the occurrence of the COVID-19 outbreak was disinfecting of hands after returning home (53.4% increase during partial lockdown and 56% increase at present compared with before the COVID-19 outbreak). Additionally, the tendency to seek ASF with approved quality checks and a clear origin had increased (6.7% and 8% increase during partial lockdown and at present, respectively), as had washing ASF carefully before cooking (6.6% and 5.3% increase during partial lockdown and at present, respectively). Consumers had a reduced frequency of eating rare or undercooked ASF (5.4% and 6.7% increase during partial lockdown and at present, respectively). However, the practices regarding frequently washing kitchen equipment after preparing ASF, frequently washing hands after preparing ASF, using separate kitchen utensils for raw and cooked ASF, and wearing gloves when preparing raw ASF remain unchanged for before the outbreak, during partial lockdown, and at present (Appendix A).

In rural areas, similarly to in U/PU areas, disinfecting of hands after returning home was the practice that was highly observed after the COVID-19 outbreak (62.7% increase during partial lockdown and 64% increased at present compared with before the COVID-19 outbreak). Consumers reduced their frequency of eating rare or undercooked ASF (9.4% and 6.7% reduction during partial lockdown and at present, respectively). Additionally, washing ASF carefully before cooking (5.3% and 4% increase during partial lockdown and at present, respectively), frequently washing kitchen equipment after preparing ASF (5.3% and 6.6% increase during partial lockdown and at present, respectively), frequently washing hands after preparing ASF (6.7% increased each of during partial lockdown and at present), and using separate kitchen utensils for raw and cooked ASF (4% increased at present) were practices that changed as a result of COVID-19. However, the COVID-19 pandemic had little effect on the practice of seeking ASF with approved quality checks and a clear origin, or on wearing gloves when preparing raw ASF (Appendix A).

#### 3.5.4. Factors Related to KAP on COVID-19 and Food Safety of Consumers

There were no significant differences in the knowledge, attitudes, and practices of consumers in U/PU and rural areas. Regarding the comparison of practices within groups, the results showed that food safety practices of the consumers during partial lockdown as well as the present time have improved from the practices at the time before the outbreak in both areas (Appendix A). However, there were no significant differences in the practices of consumers in both areas during the lockdown compared with present. 

Based on univariable analysis, 70% of the consumers overall had proper knowledge regarding COVID-19 transmission and prevention (Table 3). The results demonstrate that education levels were positively correlated with increased knowledge of COVID-19, whereby consumers with college or higher education were found to have more knowledge than those with primary education (OR = 4.24, 95%CI = 1.67, 10.79). Regarding the attitudes of consumers, the results show that about half (51%) had positive attitudes toward COVID-19 prevention and 57% had positive attitudes toward food safety practices. Compared with the lowest age group, consumers in the higher age groups were more likely to have more positive attitudes toward food safety practice (Table 3). Additionally, consumers with less than three family members in the households showed significantly higher knowledge (OR = 0.17, 95%CI = 0.05, 0.55) and more positive attitudes (OR = 0.13, 95%CI = 0.03, 0.53) toward COVID-19 than those with a higher number of family members.

In contrast to knowledge, practices toward food safety were found to be negatively correlated with education. Consumers with primary education were found to have better practices than those with higher education levels (OR = 0.29, 95%CI = 0.10, 0.86). Regarding occupations, consumers who operate their own businesses were found to have better practices toward food safety than those with other occupations (OR = 0.26, 95% CI = 0.10, 0.64).

The multivariable regression analyses demonstrated that education and the number of family members were factors that were significantly related to knowledge of COVID-19, as shown in Appendix A. Additionally, factors that were significantly related with attitudes toward COVID-19 prevention were gender, education, occupation, and number of family members, while age group and occupation were the factors that were related to attitudes toward food safety practices. Lastly, education and occupation were the influencing factors for practices toward food safety of the consumers in this study (Appendix A).

## 4. Discussion

### 4.1. Impacts of COVID-19 and KAP of ASF Retailers

Most of the retailers (above 80%) in both areas did not face difficulties in finding ASF for selling as a result of COVID-19. ASF activities were less likely to be disrupted by lockdown measures that were enacted in response to massive outbreaks [18] because animal husbandry and food from animal production for domestic consumption were conducted within the province. Thus, restrictions of movements between provinces did not affect rearing, processing, and selling activities. Additionally, there was no significant change in the number of shops in the same areas during partial lockdown compared with the time prior an outbreak and the present time. This implies that COVID-19 did not cause severe damage to ASF business in the examined areas. Regarding the frequency of selling, most retailers sold ASF as their main career activity; thus, selling occurred mostly daily in both areas. Retailers were able to maintain their businesses with the same amount of time and labor spent on selling as prior to the outbreak. Owing to less production, processing, and distribution in other places due to the first COVID-19 wave, meat and meat product prices surged due to panic buying. Later, lockdown restrictions and decreasing consumer purchasing power reduced meat output and demand, lowering meat prices [15]. Up to 70% of the retailers in U/PU and 60% in rural areas had decreased income because buyers went to markets less frequently and bought less than before, resulting in a decrease to about half of the regular income due to the impacts of COVID-19. Furthermore, less demand and purchasing power of the consumers led to lower ASF supply during partial lockdown until the time of interviewing. There were some retailers whose income increased during the COVID-19 outbreak due to increase in the number of customers purchasing their type of ASF with a cheaper price than that of other meat. Other reasons included increased demand in buying for the purpose of storing food in response to the lockdown measures, as well as support expenditure from the government to subsidize the loss of consumers’ income and to stimulate the economy in the country.

The COVID-19 pandemic has had multidimensional critical impacts on people’s lifestyles and economies worldwide. Almost 100% of the retailers (98.7% in U/PU and 100% in rural areas) in this study had heard about COVID-19, which corresponds with the results of other previously published studies [31,32]. However, approximately 60% and 70% in U/PU and rural areas, respectively, had knowledge that the causative agent is a virus. About three-quarters of the retailers had good knowledge of COVID-19 and its transmission and prevention. Additionally, the majority (80.6% and 81.9% in U/PU and rural areas, respectively) did not agree that ASF is a possible source of infection, reflected in their positive attitudes toward ASF consumption, which was consistent with a study in Kansas [27]. Regarding their knowledge and perspectives, they tended to avoid eating wildlife because of its possible risk of COVID-19 infection. Most had positive attitudes toward food safety practices and were not worried about their consumers being concerned that they might have health problems when wearing a mask or gloves while selling ASF products (38.9% in U/PU and 62.7% in rural areas). The introduction of COVID-19 has largely changed the attitudes of retailers toward personal hygiene, whereby wearing gloves and a mask when selling was considered not a proper habit in the past from the consumers’ viewpoint. 

Hand washing, raw ASF separation from other food items, cold storage of ASF, use of personal protective equipment (such as gloves and masks), using disinfectants to clean surfaces that come into contact with raw ASF, and vaccinating against COVID-19 have been recommended for workers that are handling animal products [33]. In U/PU areas, the practices of hand disinfection and wearing gloves and a mask when selling have greatly increased because of the outbreak of COVID-19, while in rural areas, the practices that have increased are hand disinfection and ASF quality checks by authorities. However, practices of wearing gloves and a mask have hardly increased in rural areas. As with the first COVID-19 wave, there was no free government-administered vaccination available at the time of the interview. The vaccines were imported for the first time in February 2021 and stored for frontline healthcare workers, individuals with comorbidities, and the elderly [34]. Vaccines were administered to the general public in the third quarter of 2021, after the outbreak had continued for over a year. Personal hygiene and disinfecting facilities were sufficient for retailers to maintain their businesses, according to the scope of the interviews. A total of 15% of retailers ceased selling ASF during the lockdown period (data not shown). Interestingly, in both areas, the retailers were still eating and drinking at their shops during all periods of observation. This may be because of the full-time nature of selling, as mentioned earlier, resulting in the routine habits of eating and drinking occurring within the shops. 

Additionally, the retailers in rural areas showed higher scores for attitudes than the U/PU group. Comparing between knowledge, attitudes, and practices during partial lockdown, the proportions of the retailer respondents that were assigned in the good category was highest in knowledge, but the proportions of positive attitudes and good practices were lower. This implies that even for the respondents that had good knowledge of COVID-19, their attitudes (toward COVID-19 prevention and food safety practices) and practices were not improved by the knowledge that they had. Additionally, practices toward food safety of the retailers had been significantly improved during the lockdown period and at present compared with before the outbreak. Interestingly, the practices at present were not statistically significantly different from during the lockdown. This illustrates that the retailers were able to improve their practices of food safety and able to maintain good practices in ASF selling until now. As the virus was designated as a biological hazard in food production, it is theoretically possible for food, particularly ASF, to be contaminated by direct contact with infected food handlers rather than animals [8]. Personal hygiene, as described in this study’s evaluation of food safety practices and as a component of good hygiene practices (GHP) in the food chain, was crucial for preventing the contamination of food by any pathogen, including COVID-19, by food business operators [35,36]. After the COVID-19 outbreak, as observed during the partial lockdown and currently, ASF retailers have paid more attention to improving their hygiene to minimize disease transmission and to ensure the hygienic quality of their products. 

In addition, results from univariate and multivariable regression analyses showed female retailers had more knowledge about COVID-19 than male retailers, in accordance with the results of other studies [22,26,32,37], while education was not significantly associated with knowledge. In contrast to studies in China and Nigeria [20,22], our study found that COVID-19 knowledge was not significantly correlated with attitudes toward COVID-19 prevention.

### 4.2. Impacts of COVID-19 and KAP of Consumers

The consumer demands for a type of ASF in each of the areas was reflected by an average amount of ASF purchasing per shopping trip and corresponded to the proportion of retailers, in which the number of fish/seafood retailers was highest in U/PU, while pork was highest in the rural areas. Regarding the location of ASF purchases, although online shopping became the preferred method after the introduction of COVID-19 [38], ASF was not a food type that consumers chose to purchase online. This may be due to the various characteristics of ASF which limited them from making decisions regarding the appearance and quality of ASF products when shopping online. Most consumers had not been affected by COVID-19 with regard to their ASF purchasing. The amount of ASF purchasing by type remained constant for the three periods of observation, except for the significantly decreased amount of pork that was purchased in rural areas during partial lockdown, which was the most preferred type of ASF in these areas. Due to a decrease in income and a desire to reduce household expenditure, consumers in rural areas may have decreased their consumption of pork and shifted to a lower-priced ASF. In Akter’s study on the effects of COVID-19-related stay-at-home restrictions on food prices in Europe, it was determined that the restrictions were significantly associated with higher prices for meat, fish and seafood, and vegetables, whereas the prices for other foods, such as milk and eggs, were unaffected [39]. A study of the impacts of COVID-19 on the global agricultural markets revealed that the COVID-19 pandemic led to a sharp decline in economic growth and a 7–18% decrease in international meat prices in 2020 as a short-term disruption [40]. Global food consumption was largely unaffected, however, due to the inelastic demand for most agricultural commodities and the short duration of the shock.

Regarding the effects of COVID-19 on consumer income, nearly all the consumers were affected by the pandemic, with their income decreasing by around half of its normal level. The majority (89.3% in U/PU and 78.7% in rural areas) had operated their own businesses as their primary job. As a result of their income losses, they most frequently noted a decline in the number of clients in their enterprises. In 2020, Thailand’s annual GDP decreased by 6.1% compared with the global GDP, which had decreased by 3.3% [41]. During the COVID-19 outbreak, foreigner travel restrictions [23] and public and business place closures were implemented to prevent the spread of disease [18], thereby reducing the number of travelers and customers to the interviewed consumers’ businesses.

The greatest threat to food security is not disruptions in supply networks, but rather the disastrous consequences of COVID-19 on livelihoods and employment [17]. Food insecurity, in terms of accessibility, was raised by the consumers as an issue that they perceived due to the COVID-19 outbreak (Appendix A). Worrying about not having enough food to eat (43% and 36% in U/PU and rural areas, respectively), being unable to eat healthy and nutritious food (27% and 13% in U/PU and rural areas, respectively), and eating only a few kinds of food (17% in each area) because of lack of money or other resources were listed as the most concerning issues for consumers during the COVID-19 outbreak. A survey from Elsahoryi’s study indicated that nearly 60% of the respondents felt moderate to severe food insecurity during the COVID-19 outbreak and subsequent quarantine. Since the COVID-19 outbreak, household food insecurity has increased by 32.3%, according to Niles’s study [42]. Additionally, low income per capita (below the poverty line), high number of family members (≥8), younger ages (18–30 years old), and living in a rented house were factors that were associated with food insecurity [43]. Corresponding to Mandal’s study, the COVID-19 impacts of reduced income and loss of jobs resulted in experiencing food insecurity, especially when the outbreak lasted for a long period [38].

About 70% of the consumers had good knowledge about the COVID-19 causative agent and its transmission. Regarding their knowledge, COVID-19 is less likely to be transmitted through consumption of ASF products. In U/PU, nearly half thought that wildlife could be a possible source of infection, while the other half thought that this is not possible. However, avoiding eating wildlife, raw products such as salad and fruit, and ASF were raised as unimportant practices with respect to consumers’ knowledge in rural areas. Regarding raw foodstuffs, specifically fruits and vegetables, the Kartari study indicated that COVID-19 promotes healthy eating habits among individuals; individuals in China, Portugal, and Turkey have increased their consumption of fresh fruits and vegetables due to the pandemic [44]. During the pandemic, the consumption of red meat remained constant among Chinese and Turkish individuals, while it decreased among Portuguese citizens [44,45,46]. The primary reason for the decline in red meat consumption was the economic status of the consumers [45]. The consumers were unaware of contracting the disease from consumption of ASF, as reflected from their attitudes. Thus, changes in ASF purchasing in terms of frequency and amount were not obviously observed in this study. In contrast, Attwood’s study on consumer perceptions of ASF suggests that the media’s emphasis on the zoonotic origin of coronaviruses could decrease the demand for meat and wildlife products [47]. Contrary to their knowledge, with the lack experience of wildlife consumption, avoiding consuming of those products was raised as a measure to prevent the disease transmission, secondary to complying with social distancing, personal hygiene, and the use of protective equipment in response to the outbreak. Despite their responses on the knowledge of preventive measures of COVID-19, only staying at home and avoiding meeting with strangers were ideal practices but were not considered applicable according to their perspectives. Regarding the attitudes toward food safety practices, consistent with the retailers’ responses, they did not feel that shop owners who used masks and gloves while selling were trying to hide health problems. According to food safety practices, hand disinfection after returning home was a practice that greatly increased due to the COVID-19 outbreak in both areas. 

Similarly, consumer practices were significantly improved during partial lockdown and at present compared with before the COVID-19 outbreak. However, comparing between the practices during lockdown and at present, the results were not significantly different, which means that consumers have maintained their practices since COVID-19 was introduced into the areas. In addition to improved personal hygiene among ASF retailers and consumers with the introduction of COVID-19 during the initial phase of the outbreak, food preparation was a crucial step to prevent the consumers from biological hazards, given that they are end users in the food chain. The World Health Organization developed the five keys to safer food in order to reduce the concepts underlying food safety into easy stages of procedures for food handlers, including consumers [48]. The five keys to safe food are keep clean, separate raw and cooked, cook thoroughly, keep food at safe temperatures, and use safe water and raw materials [48]; some of these items were evaluated in the questionnaire. Since prior to the COVID-19 outbreak, it appears that the consumers who participated in this study have maintained all of their good practices relating to food preparation, with the exception of washing their hands after returning home from going out, which has greatly improved since the introduction of COVID-19 (see Appendix A). However, it had nothing to do with meal preparation.

Consistent with other studies, education level demonstrated a positive correlation with knowledge of COVID-19 [23,24,25,26,37]. Having college or higher level of education compared with primary education showed a significant influence according to chi-square analysis. In contrast with knowledge toward COVID-19, consumers with high education were less likely to engage in practices that were related to food safety compared to those with primary education. Additionally, consumers with good knowledge of COVID-19 were less likely engage in practices that were related to food safety, which corresponds with the results for retailers. However, food safety attitudes were positively correlated with attitudes toward COVID-19 prevention, and similarly with retailers.

### 4.3. Study Strengths and Limitations

This is the first investigation into the effects and KAP of the COVID-19 outbreak on ASF retailers and consumers in Thailand. The scope of the study encompasses three distinct time periods (before the COVID-19 outbreak, during the partial lockdown, and at the time of the interviews) in order to examine changes in livelihood and practices occurring as a result of the outbreak. In addition, because of the study’s food-related population, the assessment of food safety practices was also conducted to investigate the relationship between disease outbreaks and the improvement of food safety practices. This study did not, however, gather information regarding the respondents’ sources of knowledge regarding COVID-19 transmission and prevention and direct practices/responses for COVID-19 prevention. Lastly, limitations may exist when comparing KAP across studies due to the different question sets that are used in each study. 

## 5. Conclusions

Overall, the COVID-19 pandemic did not affect ASF selling and purchasing activities. However, considerable income losses were reported among retailers and consumers in both U/PU and rural areas. The incomes of around 76% of U/PU and 65% of rural retailers were halved during the partial lockdown, while the incomes of around 44% of U/PU and 60% of rural consumers were halved. In addition, food insecurity was raised as a consumer concern due to the economic effects of COVID-19. Both retailers and consumers demonstrated modifications in their food safety practices. During the partial lockdown, remarkable improvements in food safety practices were observed compared with the time preceding the pandemic. COVID-19 has had positive effects on the food safety practices of respondents, as they have maintained their good practices to the present day. As the virus was designated as a biological hazard in food production, this study presented results on the practices of respondents in the context of food safety to avoid disease transmission and infection. Our findings provide information for policy-makers and stakeholders to analyze the effects of COVID-19 on Thai ASF retailers and consumers, as well as the parameters that are related with their KAP for disease prevention and food safety.

## Figures and Tables

**Figure 1 ijerph-19-10187-f001:**
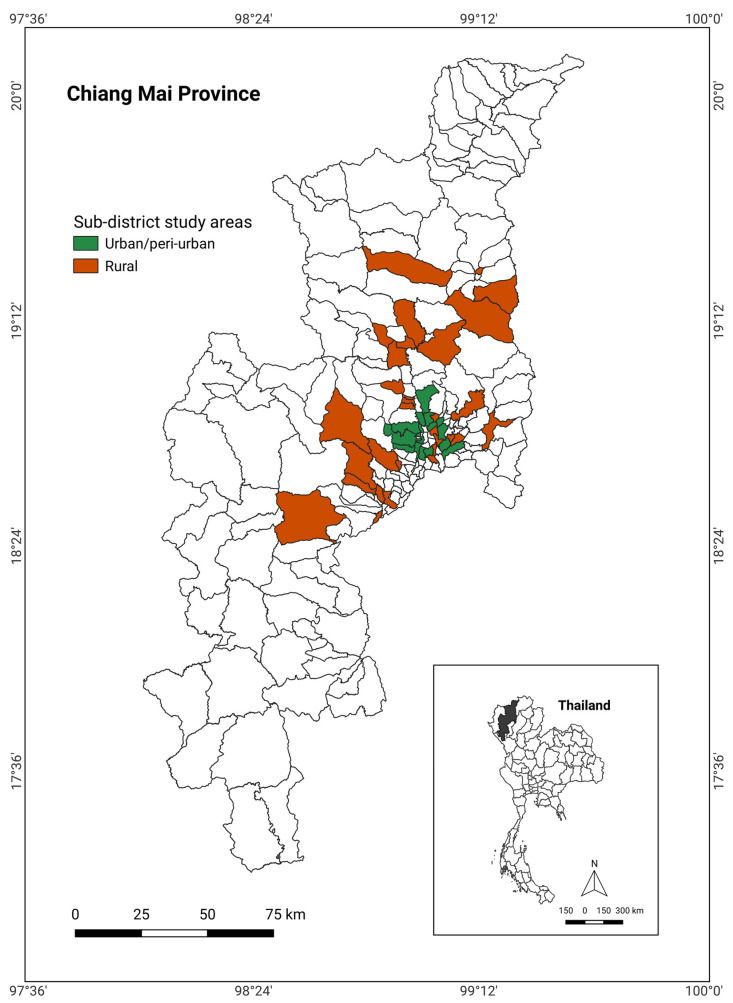
Subdistrict administrative map of Chiang Mai province.

**Figure 2 ijerph-19-10187-f002:**
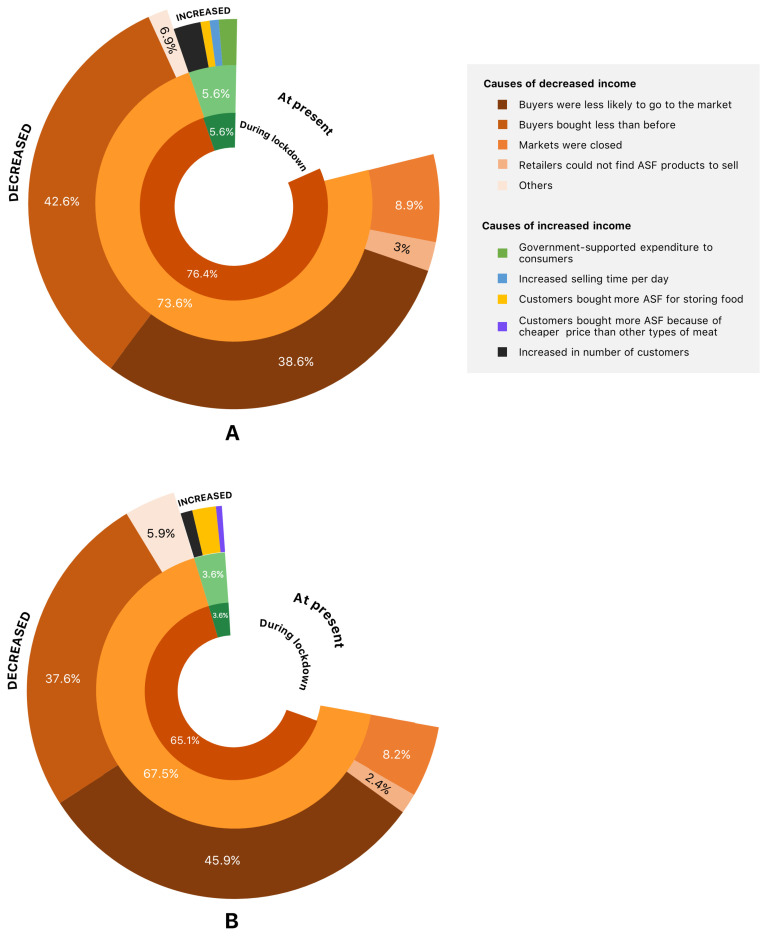
Impacts of COVID-19 on ASF selling income of the retailers in (**A**) U/PU and (**B**) rural areas. The outermost chart details the factors that contributed to decreased/increased income, and the percentage for each item was computed using only retailers that were affected by the COVID-19 pandemic (100% in total).

**Figure 3 ijerph-19-10187-f003:**
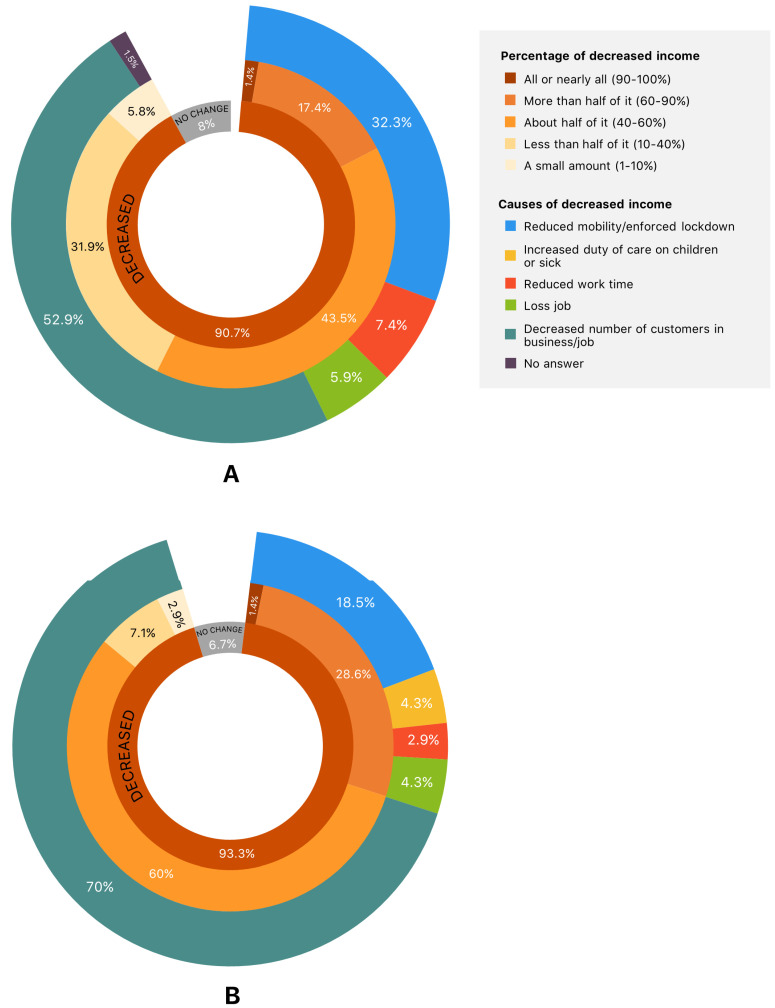
Impacts of COVID-19 on consumers’ incomes in (**A**) U/PU and (**B**) rural areas. The center and outermost charts detail the quantity and causes of income loss, respectively. The percentage for each item was computed using only retailers whose income had decreased (100% in total).

**Table 1 ijerph-19-10187-t001:** Demographic characteristics of animal source food (ASF) retailers and consumers.

Item	No. of Retailers, *n* (%)		No. of Consumers, *n* (%)	
	Urban/Peri-Urban (*n* = 72)	Rural (*n* = 83)	Urban/Peri-Urban (*n* = 75)	Rural (*n* = 75)
**Gender**				
Female	52 (72.2)	57 (68.7)	64 (85.3)	60 (80.0)
Male	20 (27.8)	26 (31.3)	11 (14.7)	15 (20.0)
**Age (mean ± SD)**	**46.8 ± 12.9**	**48.0 ± 14.6**	**44.9 ± 13.7**	**50.9 ± 12.7**
<20	0	0	1 (1.3)	0
20–29	7 (9.7)	13 (15.7)	12 (16.0)	5 (6.7)
30–39	17 (23.6)	11 (13.3)	12 (16.0)	12 (16.0)
40–49	17 (23.6)	18 (21.7)	24 (32.0)	12 (16.0)
50–60	19 (26.4)	23 (27.7)	14 (18.7)	27 (36.0)
>60	12 (16.7)	18 (21.7)	12 (16.0)	19 (25.3)
**Education**				
Illiterate	3 (4.2)	8 (9.6)	1 (1.3)	2 (2.7)
Primary school	29 (40.3)	34 (41.0)	17 (22.7)	36 (48.0)
Secondary school	3 (4.2)	7 (8.4)	12 (16.0)	8 (10.7)
High school	14 (19.4)	17 (20.5)	16 (21.3)	12 (16.0)
College or higher	23 (31.9)	17 (20.5)	29 (38.7)	17 (22.7)
**Type of ASF retailer**				
Pork	20 (27.8)	36 (43.4)	NA	NA
Poultry	17 (23.6)	14 (16.9)		
Beef	4 (5.6)	9 (10.8)		
Fish/seafood	31 (43.1)	24 (28.9)		
**Business type**				
Wholesale only	1 (1.4)	0	NA	NA
Retail only	53 (73.6)	71 (85.5)		
Wholesale and retail	18 (25.0)	12 (14.5)		
**Occupation**				
Farmer	NA	NA	0	2 (2.7)
Government officer/staff	NA	NA	2 (2.7)	2 (2.7)
Worker	NA	NA	1 (1.3)	7 (9.3)
Private business	72 (100)	83 (100)	67 (89.3)	59 (78.7)
Housewife	NA	NA	3 (4.0)	4 (5.3)
Students	NA	NA	2 (2.7)	1 (1.3)
**Household monthly income**				
≤200 USD	NA	NA	3 (4.0)	8 (10.7)
201–400 USD			21 (28.0)	18 (24.0)
401–600 USD			12 (16.0)	18 (24.0)
601–800 USD			16 (21.3)	13 (17.3)
801–1000 USD			8 (10.7)	9 (12.0)
1001–2000 USD			6 (8.0)	5 (6.7)
>2000 USD			7 (9.3)	1 (1.3)
NA *			2 (2.7)	3 (4.0)
**Number of family members**				
<3	NA	NA	29 (38.7)	23 (30.7)
3–5			38 (50.7)	43 (57.3)
>5			8 (10.7)	9 (12.0)

* Not available.

**Table 2 ijerph-19-10187-t002:** Univariable analysis on the knowledge, attitudes, and practices of retailers.

Variable	*n*	Knowledge toward COVID-19		Attitudes toward COVID-19 Prevention		Attitudes toward Food Safety Practices		Practices toward Food Safety (During Partial Lockdown)	
		Good *n* (%)	OR_crude_ (95%CI)	Good *n* (%)	OR_crude_ (95%CI)	Good *n* (%)	OR_crude_ (95%CI)	Good *n* (%)	OR_crude_ (95%CI)
**Gender**									
Male	46	28 (60.9)	1	35 (76.1)	1	25 (54.3)	1	22 (47.8)	1
Female	109	88 (81.6)	**2.69 ^b^ (1.26, 5.76)**	69 (73.1)	0.54 (0.25, 1.18)	66 (60.6)	1.29 (0.64, 2.59)	64 (58.7)	1.55 (0.78, 3.10)
**Age group**									
20–29	20	18 (90.0)	1	13 (65.0)	1	10 (50.0)	1	8 (40.0)	1
30–39	28	20 (71.4)	0.28 (0.05, 1.48)	19 (67.9)	1.14 (0.34, 3.83)	15 (53.6)	1.15 (0.37, 3.64)	20 (71.4)	**3.75 (1.11, 12.62)**
40–49	35	27 (77.1)	0.38 (0.07, 1.97)	23 (65.7)	1.03 (0.33, 3.27)	21 (60.0)	1.50 (0.50, 4.54)	17 (48.6)	1.42 (0.47, 4.31)
50–60	42	31 (73.8)	0.31 (0.06, 1.57)	27 (64.3)	0.97 (0.32, 2.95)	25 (59.5)	1.47 (0.50, 4.29)	27 (64.3)	2.70 (0.90, 8.07)
>60	30	20 (66.7)	0.22 (0.04, 1.15)	22 (73.3)	1.48 (0.44, 5.04)	20 (66.7)	2.00 (0.63, 6.38)	14 (46.7)	1.31 (0.42, 4.13)
**Education**									
Illiterate	11	7 (63.6)	1	6 (54.5)	1	6 (54.5)	1	4 (36.4)	1
Primary school	63	41 (65.1)	1.06 (0.28, 4.04)	43 (68.3)	1.79 (0.49, 6.57)	38 (60.3)	1.27 (0.35, 4.60)	38 (60.3)	2.66 (0.70, 10.04)
Secondary school	10	8 (80.0)	2.29 (0.32, 16.51)	8 (80.0)	3.33 (0.47, 23.47)	6 (60.0)	1.25 (0.22, 7.08)	6 (60.0)	2.63 (0.45, 15.31)
High school	31	25 (80.6)	2.38 (0.52, 10.86)	21 (67.7)	1.75 (0.43, 7.14)	17 (54.8)	1.01 (0.25, 4.03)	20 (64.5)	3.18 (0.76, 13.32)
College or higher	40	35 (87.5)	4.00 (0.85, 18.75)	26 (65.0)	1.55 (0.40, 5.99)	24 (60.0)	1.25 (0.33, 4.80)	18 (45.0)	1.43 (0.36, 5.68)
**Type of ASF**									
Pork	56	38 (67.9)	1	41 (73.2)	1	35 (62.5)	1	40 (71.4)	1
Poultry	31	23 (74.2)	1.36 (0.51, 3.63)	19 (61.3)	0.58 (0.23, 1.47)	17 (54.8)	0.73 (0.30, 1.78)	16 (51.6)	0.43 (0.17, 1.06)
Beef	13	11 (84.6)	2.61 (0.52, 13.00)	9 (69.2)	0.82 (0.22, 3.08)	6 (46.2)	0.51 (0.15, 1.74)	7 (53.8)	0.47 (0.14, 1.61)
Fish/seafood	55	44 (80.0)	1.89 (0.80, 4.51)	35 (63.6)	0.64 (0.29, 1.44)	33 (60.0)	0.90 (0.42, 1.93)	23 (41.8)	**0.29 (0.13, 0.63)**
**Business type ^a^**									
Retail only	124	89 (71.8)	1	86 (69.4)	1	81 (65.3)	1	71 (57.3)	1
Wholesale and retail	30	26 (86.7)	2.56 (0.83, 7.86)	18 (60.0)	0.66 (0.29, 1.51)	9 (30.0)	**0.23 (0.10, 0.54)**	15 (50.0)	0.75 (0.34, 1.66)
**COVID-19 knowledge**									
Poor knowledge	39	0 (0)	-	23 (59.0)	1	24 (61.5)	1	26 (66.7)	1
Good knowledge	116	116 (100)	-	81 (69.8)	1.61 (0.76, 3.41)	67 (57.8)	0.85 (0.41, 1.80)	60 (51.7)	0.54 (0.25, 1.14)
**Food safety attitudes**									
Poor attitude	64	49 (76.6)	1	28 (43.8)	1	0 (0)	-	35 (54.7)	1
Positive attitude	91	67 (73.6)	0.85 (0.41, 1.80)	76 (83.5)	**6.51 (3.10, 13.68)**	91 (100)	-	51 (56.0)	1.06 (0.56, 2.01)

OR, odds ratio; CI, confidence interval. Value of 1 was the category used as a reference for the comparable category. ^a^ Wholesale business type (*n* = 1) was excluded from the analysis. ^b^ Values in bold were indicated significantly difference with the reference category (*p*-value < 0.05).

**Table 3 ijerph-19-10187-t003:** Univariable analysis of the knowledge, attitudes, and practices of consumers.

Variable	n	Knowledge toward COVID-19		Attitudes toward COVID-19 Prevention		Attitudes toward Food Safety Practices		Practices toward Food Safety (During Partial Lockdown)	
		Good n (%)	ORcrude (95%CI)	Good n (%)	ORcrude (95%CI)	Good n (%)	ORcrude (95%CI)	Good n (%)	ORcrude (95%CI)
**Gender**									
Male	26	17 (65.4)	1	9 (34.6)	1	12 (46.2)	1	16 (61.5)	1
Female	124	88 (71.0)	1.29 (0.53, 3.17)	67 (54.0)	2.22 (0.92, 5.36)	73 (58.9)	1.67 (0.71, 3.91)	81 (65.3)	1.18 (0.49, 2.82)
**Age group ^a^**									
20–29	17	13 (76.5)	1	8 (47.1)	1	4 (23.5)	1	11 (64.7)	1
30–39	24	18 (75.0)	0.92 (0.22, 3.94)	13 (54.2)	1.33 (0.38, 4.62)	15 (62.5)	**5.42 ^d^ (1.35, 21.80)**	17 (70.8)	1.32 (0.35, 5.00)
40–49	36	31 (86.1)	1.91 (0.44, 8.26)	17 (47.2)	1.01 (0.32, 3.20)	20 (55.6)	**4.06 (1.11, 14.90)**	22 (61.1)	0.86 (0.26, 2.84)
50–60	41	24 (58.5)	0.43 (0.12, 1.56)	23 (56.1)	1.44 (0.46, 4.47)	28 (68.3)	**7.00 (1.91, 25.67)**	27 (65.9)	1.05 (0.32, 3.44)
>60	31	18 (58.1)	0.43 (0.11, 1.61)	15 (48.4)	1.05 (0.32, 3.45)	18 (58.1)	**4.50 (1.19, 16.99)**	20 (64.5)	0.99 (0.29, 3.42)
**Education ^b^**									
Primary school	53	28 (52.8)	1	31 (58.5)	1	32 (60.4)	1	39 (73.6)	1
Secondary school	20	16 (80.0)	**3.57 (1.05, 12.11)**	9 (45.0)	0.58 (0.21, 1.64)	10 (50.0)	0.66 (0.23, 1.85)	9 (45.0)	**0.29 (0.10, 0.86)**
High school	28	21 (75.0)	2.68 (0.97, 7.36)	14 (50.0)	0.71 (0.28, 1.78)	17 (60.7)	1.01 (0.40, 2.59)	17 (60.7)	0.55 (0.21, 1.47)
College or higher	46	38 (82.6)	**4.24 (1.67, 10.79)**	20 (43.5)	0.55 (0.25, 1.21)	24 (52.2)	0.72 (0.32, 1.59)	30 (65.2)	0.67 (0.28, 1.59)
**Occupation**									
Private business	126	87 (69.0)	1	72 (57.1)	1	78 (61.9)	1	88 (69.8)	1
Others	24	18 (75.0)	1.34 (0.50, 3.65)	4 (16.7)	**0.15 (0.05, 0.46)**	7 (29.2)	**0.25 (0.10, 0.66)**	9 (37.5)	**0.26 (0.10, 0.64)**
**Household monthly** **income (USD) ^c^**									
≤200	11	7 (63.6)	1	5 (45.5)	1	5 (45.5)	1	7 (63.6)	1
201–400	39	25 (64.1)	1.02 (0.25, 4.10)	16 (41.0)	0.83 (0.22, 3.21)	19 (48.7)	1.14 (0.30, 4.37)	25 (64.1)	1.02 (0.25, 4.10)
401–600	30	21 (70.0)	1.33 (0.31, 5.72)	15 (50.0)	1.20 (0.30, 4.80)	19 (63.3)	2.07 (0.51, 8.41)	17 (56.7)	0.75 (0.18, 3.11)
601–800	29	23 (79.3)	2.19 (0.48, 10.04)	17 (58.6)	1.70 (0.42, 6.88)	22 (75.9)	3.77 (0.88, 16.24)	19 (65.5)	1.09 (0.26, 4.62)
801–1000	17	12 (70.6)	1.37 (0.27, 6.87)	12 (70.6)	2.88 (0.59, 13.98)	9 (52.9)	1.35 (0.29, 6.18)	13 (76.5)	1.86 (0.35, 9.79)
1001–2000	11	8 (72.7)	1.52 (0.25, 9.29)	4 (36.4)	0.69 (0.12, 3.78)	4 (36.4)	0.69 (0.12, 3.78)	7 (63.6)	1.00 (0.18, 5.68)
>2000	8	6 (75.0)	1.71 (0.23, 12.89)	5 (62.5)	2.00 (0.31, 12.84)	6 (75.0)	3.60 (0.49, 26.40)	7 (87.5)	4.00 (0.35, 45.38)
**Number of family** **members**									
<3	52	42 (80.8)	1	32 (61.5)	1	33 (63.5)	1	34 (65.4)	**1**
3–5	81	56 (69.1)	0.53 (0.23, 1.23)	41 (50.6)	0.64 (0.32, 1.30)	45 (55.6)	0.72 (0.35, 1.47)	50 (61.7)	0.85 (0.41, 1.76)
>5	17	7 (41.2)	**0.17 (0.05, 0.55)**	3 (17.6)	**0.13 (0.03, 0.53)**	7 (41.2)	0.40 (0.13, 1.23)	13 (76.5)	1.72 (0.49, 6.05)
**COVID-19 knowledge**									
Poor knowledge	45	0 (0)	-	19 (42.2)	1	24 (53.3)	1	35 (77.8)	1
Good knowledge	105	105 (100)	-	57 (54.3)	1.63 (0.80, 3.29)	61 (58.1)	1.21 (0.60, 2.45)	62 (59.0)	**0.41 (0.18, 0.92)**
**Food safety attitudes**									
Poor attitude	65	44 (67.7)	1	23 (35.4)	1	0	-	37 (56.9)	1
Positive attitude	85	61 (71.8)	1.21 (0.60, 2.45)	53 (62.4)	**3.02 (1.55, 5.92)**	85 (100)	-	60 (70.6)	1.82 (0.92, 3.58)

OR, odds ratio; CI, confidence interval. Value of 1 was the category used as a reference for the comparable category. ^a^ Age group of less than 20 (*n* = 1), ^b^ illiterate (*n* = 3), and ^c^ data not available on household monthly income (*n* = 5) were excluded from the analysis. ^d^ Values in bold were indicated significantly difference with the reference category (*p*-value < 0.05).

## Data Availability

Not applicable.

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
