# Peer review of "Impacts of the Pandemic, Animal Source Food Retailers’ and Consumers’ Knowledge and Attitudes toward COVID-19, and Their Food Safety Practices in Chiang Mai, Thailand"

_ijerph, 2022, doi:10.3390/ijerph191610187_

Round 1

Reviewer 1 Report

Dear authors, 

despite appreciating the originality of the topic, it is not clear from the manuscript the rationale behind it and why the Knowledge toward COVID-19 should inflence the food safety or the food consumption. 

It looks like these two topics are not strectly related so my suggestion is to keep them separated and to proceed to two different in depth analysis, therefore in two different pubblications. Some findings could be more related with food insecurity than Covid-19 attitude and knowledge. 

Author Response

Dear Reviewer,

We appreciate you dedicating your time for reading and commenting on our work. Now, we complete responding all of your issues/queries. Please see an attachment for our responses.

Best Regards,

Reviewer 2 Report

The paper submitted deals with the effects of COVID-19 on consumer knowledge, attitude and practice related to animal source food.
The topic discussed in the paper, although of interest as is the first indication for consumer attitudes in the geographic area studied, in the opinion of this reviewer may be seen as partially fitting into the journal scope, as the health issues related with consumer purchasing attitude are somehow overshadowed by the economic aspects involved in the survey.

Apart from the above considerations, references appear adequate for the study conducted, as the methodology used. Some considerations follow:

- in the introduction, in addition to the mere description of health impact of the COVID-19 pandemic, the authors should dwell more on the consequences of limitation of movement by the authorities, to better frame the study conducted, with addition of appropriate bibliographic references.

- in the materials and methods section, it is appropriate to specify the reasons that led to the choice of the sub-district areas studied.

This reviewer does not find other elements that make the submitted paper not suitable for publication, although thinks that its scope does not fully reflect the editorial line of the journal. Therefore, it is recommendable a minor revision to focus more on the health aspects obtained from the survey.

Author Response

(The authors gave the same response as above.)

Reviewer 3 Report

This study aims to assess the impact of COVID-19 on meat supply and consumption as well as retailers and consumers KAP regarding COVID-19 and food safety. It is a well written and discussed paper. Following are some general comments to make the paper more appealing to readers.

1- the title do not reflect the content of the paper. Revise it.

2- the introduction is clear and coherent; however, I would expect more information about why the study is targeting animal food" would be also interesting to highlight more the impact of Covid 19 on the supply chain, demand, consumer behavior worldwide. Alot of data are published recently in peer review journals and worth presenting. Maybe talk less about covid 19 and more on how this have affected KAP , food safety and the relationship between all. Give rational behind why it was conducted in Thailand, why it was conducted in 2 areas, why there is a need to assess retailers and consumers in one study....

3- materials and methods well described; however, you didn't talk about the questionnaire (number of questions or sections, types of questions) source of the questions or references.

4- results - excellent interpretation. Would you consider deciding the paper into 2 different ones. The first tackling consumption and KAP related to covid and second kap of food safety.

5- discussion it would be interesting to discuss further the effect of food safety and bring some relevant results or show the relationship between COVID-19 and food safety Knowledge.

 I wanted to learn more about food safety it was not enough.

Great job 

Author Response

Dear Reviewer,

We appreciate you dedicating your time for reading and commenting on our work. Now, we complete responding all of your issues/queries. Please see an attachment for our responses.

Best Regards,

This manuscript is a resubmission of an earlier submission. The following is a list of the peer review reports and author responses from that submission.